# HIV-1 Vpu is a potent transcriptional suppressor of NF-κB-elicited antiviral immune responses

Simon Langer[1,2†], Christian Hammer[3,4†‡], Kristina Hopfensperger[1†], Lukas Klein[1], Dominik Hotter[1], Paul D De Jesus[2], Kristina M Herbert[2], Lars Pache[2], Nikaïa Smith[1], Johannes A van der Merwe[1], Sumit K Chanda[2], Jacques Fellay[3,4], Frank Kirchhoff[1], Daniel Sauter[1]*

[1]Institute of Molecular Virology, Ulm University Medical Center, Ulm, Germany; [2]Infectious and Inflammatory Disease Center, Sanford Burnham Prebys Medical Discovery Institute, La Jolla, California, United States; [3]School of Life Sciences, École Polytechnique Fédérale de Lausanne, Lausanne, Switzerland; [4]Swiss Institute of Bioinformatics, Lausanne, Switzerland

**Abstract** Many viral pathogens target innate sensing cascades and/or cellular transcription factors to suppress antiviral immune responses. Here, we show that the accessory viral protein U (Vpu) of HIV-1 exerts broad immunosuppressive effects by inhibiting activation of the transcription factor NF-κB. Global transcriptional profiling of infected CD4 +T cells revealed that *vpu*-deficient HIV-1 strains induce substantially stronger immune responses than the respective wild type viruses. Gene set enrichment analyses and cytokine arrays showed that Vpu suppresses the expression of NF-κB targets including interferons and restriction factors. Mutational analyses demonstrated that this immunosuppressive activity of Vpu is independent of its ability to counteract the restriction factor and innate sensor tetherin. However, Vpu-mediated inhibition of immune activation required an arginine residue in the cytoplasmic domain that is critical for blocking NF-κB signaling downstream of tetherin. In summary, our findings demonstrate that HIV-1 Vpu potently suppresses NF-κB-elicited antiviral immune responses at the transcriptional level.
DOI: https://doi.org/10.7554/eLife.41930.001

*For correspondence:
daniel.sauter@uni-ulm.de

[†]These authors contributed equally to this work

Present address: [‡]Department of Cancer Immunology, Genentech, South San Francisco, United States

Competing interests: The authors declare that no competing interests exist.

## Introduction

Efficient replication of human immunodeficiency virus (HIV) depends on its ability to counteract or evade a variety of antiviral defense mechanisms. Both adaptive and innate immune responses exert tremendous selection pressure on the virus and form an environment that is hostile to viral replication. Nevertheless, HIV replicates and spreads successfully in its human host, still infecting about 2 million individuals each year. The success of HIV-1 in part depends on four so-called accessory proteins (Vif, Nef, Vpr, and Vpu). These viral proteins can be dispensable for viral replication in vitro, but are essential for high viral loads and efficient spread in vivo since they antagonize a variety of innate immune sensors and/or effectors. The host restriction factors APOBEC3G, tetherin, and SERINC5, for example are counteracted by Vif, Vpu, and Nef, respectively (*Chemudupati et al., 2019*; *Sauter and Kirchhoff, 2018*; *Simon et al., 2015*).

Accumulating evidence suggests that lentiviral accessory proteins not only target specific components of the cellular immune response at the protein level, but exert broad inhibitory effects by suppressing antiviral immune responses also at the transcriptional level (*Faust et al., 2017*; *Sauter and Kirchhoff, 2018*). For example, Vif, Vpr, and Vpu have all been suggested to inhibit expression of interferon-β (IFN-β) and potentially other antiviral genes by depleting the cellular transcription factor

**eLife digest** The Human Immunodeficiency Virus (or HIV for short) has infected more than 70 million people worldwide. Although effective therapies exist to prevent the replication of the virus and the development to AIDS, there is no cure or vaccine, and the virus still spreads efficiently in human populations, infecting about 1.8 million new people every year.

The unfortunate success of HIV can in part be explained by several viral proteins that trick our immune system and enable the virus to persist at high levels in the human body. For example, an HIV protein called viral protein U (Vpu) prevents infected cells from producing alarm signals such as interferons, which usually help healthy, uninfected cells to defend themselves against viruses. However, the extent to which Vpu interferes with interferons and other proteins involved in immune responses has remained unclear.

To address this question, Langer, Hammer, Hopfensperger et al. compared how different variants of HIV affect immune responses in human cells. The experiments showed that cells infected with HIV variants lacking Vpu released larger amounts of interferons and other cellular proteins that are involved in immune responses compared to HIV variants with Vpu. Further experiments showed that Vpu works by inhibiting the activation of a protein called NF-κB, which usually switches on genes that encode interferons and many other proteins involved in immune responses.

These findings demonstrate that Vpu has a broader impact on the human immune response than previously thought. In order to multiply efficiently, HIV initially requires the NF-κB protein to be active. Therefore, when NF-κB is inactive, HIV may adopt a dormant state that prevents current antiviral drug treatments from eradicating the virus in the human body. In the future, developing new drugs that can activate dormant HIV particles may therefore have the potential to help cure HIV infections.

DOI: https://doi.org/10.7554/eLife.41930.002

IRF-3 (*Doehle et al., 2012*; *Okumura et al., 2008*; *Park et al., 2014*). Vpu-mediated degradation of IRF-3, however, has been challenged and the immunosuppressive activity of Vpu has been ascribed to its ability to inhibit NF-κB activation instead (*Hotter et al., 2013*; *Manganaro et al., 2015*; *Sauter et al., 2015*). NF-κB is expressed in almost all cell types and governs the expression of hundreds of genes in response to infection. Well-described targets of NF-κB comprise pro-inflammatory cytokines such as IL-6, CXCL10, or IFN-β, as well as antiviral restriction factors including IFIT1 or ISG15 (*Hiscott et al., 1989*; *Libermann and Baltimore, 1990*; *Ohmori and Hamilton, 1993*; *Pfeffer et al., 2004*). Nevertheless, NF-κB-mediated immune activation fails to prevent HIV-1 infection. On the contrary, NF-κB activation and subsequent cytokine release may even contribute to a detrimental chronic hyper-activation of the immune system that drives disease progression in HIV-infected individuals, even under effective antiretroviral therapy (*Miedema et al., 2013*). Furthermore, NF-κB binding sites are also present in the lentiviral long terminal repeat (LTR) promoter and are essential for efficient HIV transcription and replication (*Calman et al., 1988*; *Chan and Greene, 2012*). Thus, a better understanding of how HIV-1 infection modulates NF-κB-dependent immune activation will provide important insights into the pathogenesis of AIDS and might help improve current therapeutic approaches.

Vpu has been shown to suppress NF-κB-dependent gene expression via two independent mechanisms. First, Vpu of pandemic HIV-1 group M viruses counteracts tetherin, which not only inhibits virion release, but also acts as an innate sensor activating canonical NF-κB signaling upon detection of viral infection (*Galão et al., 2012*; *Sauter et al., 2013*; *Tokarev et al., 2013*). Second, Vpu stabilizes IκB, the inhibitor of NF-κB, and prevents the nuclear translocation of NF-κB independently of tetherin (*Akari et al., 2001*; *Bour et al., 2001*; *Leulier et al., 2003*; *Sauter et al., 2015*). However, despite the key role of NF-κB in immune activation, the effects of Vpu and the relative contribution of both mechanisms on global cellular gene expression and immune activation remained unclear.

Here, we performed RNA-Seq analyses of human CD4 +T cells infected with primary HIV-1 clones containing intact, disrupted, or selectively mutated *vpu* genes. Transcription factor network analyses revealed that Vpu preferentially suppresses the expression of NF-κB target genes involved in adaptive and innate immune responses, such as antigen processing, major histocompatibility complex I

(MHC-I) presentation, type I IFN signaling or DNA/RNA sensing. Furthermore, Vpu suppressed expression of cellular host restriction factors (e.g. IFIT1-3, ISG15) and reduced the expression and release of IFNs and other pro-inflammatory cytokines (e.g. IL-6, CXCL10) from HIV-1 infected cells. In contrast to previous findings, we found no evidence for Vpu-mediated inhibition of IRF3-driven gene expression. Our results rather corroborate the hypothesis that Vpu suppresses antiviral gene expression by inhibiting the activation of NF-κB. Mutational analyses revealed that inhibition of NF-κB and the immunosuppressive effects of Vpu depend on an arginine residue in its first cytoplasmic alpha-helix, while its ability to counteract the host restriction factor and innate sensor tetherin is dispensable. In summary, our results provide new insights into the transcriptional regulation of antiviral immune responses by HIV-1 and demonstrate that the viral protein U exerts broader immunosuppressive effects than previously known.

## Results

### Generation of selective Vpu mutants

To determine the effects of Vpu-mediated tetherin counteraction and downstream inhibition of NF-κB signaling on immune activation, we generated HIV-1 mutants selectively impaired in either of these inhibitory activities (*Figure 1A*). We selected the three primary viral isolates CH293, CH077, and STCO1 since they are derived from the most prevalent HIV-1 subtypes B and C and represent different stages of infection (transmitted/founder or chronic viruses), different tropisms (R5/X4- or R5-tropic), and different risk factors (homo- or heterosexual) (*Figure 1B* and *Figure 1—figure supplement 1A*). In order to abrogate IκB stabilization and NF-κB inhibition downstream of tetherin, a previously described cytoplasmic arginine residue within Vpu was mutated to lysine (R45K in subtype B, R50K in subtype C) (*Pickering et al., 2014*; *Sauter et al., 2015*; *Yamada et al., 2018*). As expected, a luciferase-based reporter assay showed that HIV-1 constructs lacking Vpu or expressing the R/K mutant Vpu induced significantly higher levels of NF-κB activation than the respective wild type (wt) viruses (*Figure 1C*). These effects were independent of tetherin since tetherin is not expressed in HEK293T cells used in this experimental setup. Comparison with fully Vpu-deficient mutants (*vpu* stop) revealed that loss-of-function in the R/K mutants was complete for CH293 and STCO1, but only partial for CH077. Immunofluorescence microscopy showed that Vpu-mediated suppression of NF-κB activity was associated with reduced nuclear translocation of p65 (*Figure 1—figure supplement 1B*). In agreement with published data (*Kmiec et al., 2016*; *Vigan and Neil, 2010*), mutations in an alanine interface in the transmembrane domain of Vpu (A15L/A19L in subtype B, A20L/A24L in subtype C) abrogated the ability of all three viruses to decrease tetherin surface levels (*Figure 1D* and *Figure 1—figure supplement 2A*) and to counteract tetherin-mediated restriction of virus release (*Figure 1E* and *Figure 1—figure supplement 2B*). However, the AA/LL mutations had no effect on tetherin-independent NF-κB activation (*Figure 1C*). Vice versa, the R/K mutations had no significant effect on Vpu-mediated tetherin counteraction (*Figure 1D and E* and *Figure 1—figure supplement 2*). In agreement with their selective phenotype, the AA/LL and R/K mutants were expressed as efficiently as wild type Vpu (*Figure 1F*). Thus, the phenotypic properties of these viruses allowed us to examine the relative contribution of tetherin-dependent and -independent inhibition of NF-κB activation to Vpu-mediated effects on cellular gene expression and the induction of antiviral immune responses.

### Modulation of CD4 +T cell RNA expression profiles by wild type HIV-1 infection

To determine the effects of Vpu on host gene expression in a broad and unbiased manner, we analyzed the transcriptome of primary CD4 +T cells infected with the wt and the *vpu*-mutated viruses described above. Three days post-infection, RNA sequencing (RNA-Seq) was performed to enable rapid and deep profiling of transcripts. The sequencing of infected cells from four healthy donors (two women, two men) (*Figure 2A* and *Figure 2—figure supplement 1A*, *Supplementary file 1*) yielded 24.6 to 51.9 million reads per sample and an average sequence coverage of 82.6% (*Figure 2—figure supplement 1B*). No sample outliers were detected based on principal component analysis and considering Cook's distance, which measures the influence on fitted coefficients for a gene by a single sample (*Figure 2—figure supplement 1C*).

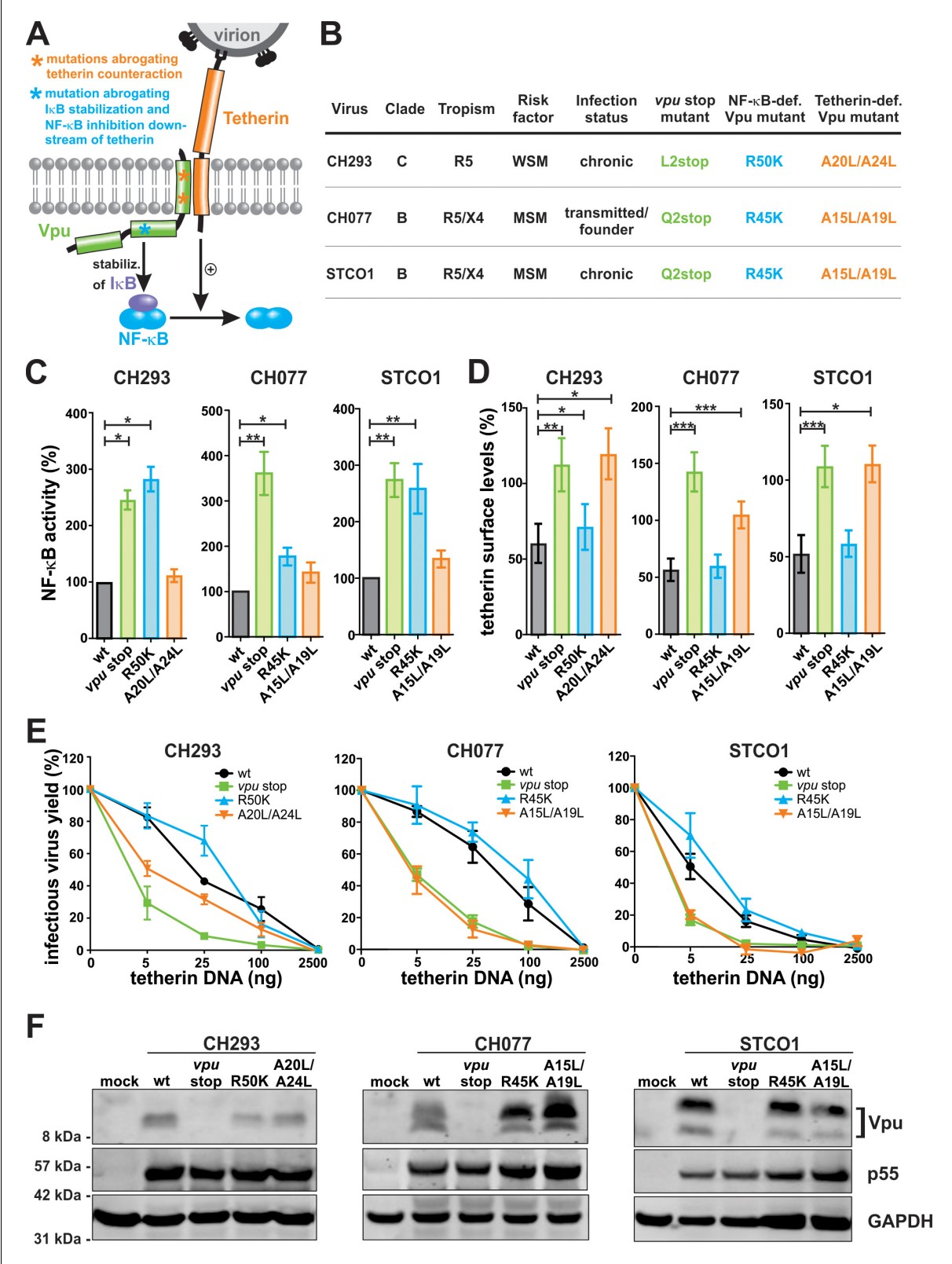

**Figure 1.** Generation of Vpu mutants that fail to inhibit NF-κB activation or to counteract tetherin. (**A**) Vpu-mediated inhibition of NF-κB activation via two independent mechanisms. Asterisks illustrate mutations in Vpu that were introduced to selectively abrogate tetherin counteraction (orange) or inhibition of NF-κB activation downstream of tetherin (blue). (**B**) Wt and mutant HIV-1 clones used in this study. MSM, man having sex with men; WSM, woman having sex with men. (**C**) Vpu-mediated inhibition of NF-κB activation. HEK293T cells were co-transfected with the indicated proviral constructs,

*Figure 1 continued on next page*

*Figure 1 continued*

a firefly luciferase-based NF-κB reporter vector, a *Gaussia* luciferase construct for normalization, and an expression vector for a constitutively active mutant of IKKβ as NF-κB inducer. Two days post-transfection, luciferase activity was determined. Mean values of three to seven independent experiments, each performed in triplicate ±SEM are shown (*p<0.05; **p<0.01; RM one-way ANOVA with Greenhouse-Geisser correction and Dunnett's multiple comparison test). (D) Vpu-mediated down-modulation of tetherin. Human PBMCs were infected with the indicated VSV-G pseudotyped HIV-1 strains. Three days post-infection, tetherin surface levels of p24 positive cells were determined by flow cytometry. Mean values of three to five independent experiments ± SEM are shown (*p<0.05; **p<0.01; ***p<0.001; RM one-way ANOVA with Greenhouse-Geisser correction and Dunnett's multiple comparison test). (E) Vpu-mediated enhancement of infectious virus yield. HEK293T cells were co-transfected with the indicated proviral constructs and increasing amounts of an expression plasmid for human tetherin. Two days post-transfection, infectious virus yield was determined by infection of TZM-bl reporter cells. Mean values of three to four independent experiments ± SEM are shown. (F) Expression of Vpu. HEK293T cells were transfected with the indicated proviral constructs. Two days post-transfection, cells were harvested and analyzed by Western Blotting.

DOI: https://doi.org/10.7554/eLife.41930.003

The following figure supplements are available for figure 1:

**Figure supplement 1.** Vpu alignment and Vpu-mediated inhibition of nuclear translocation of p65.

DOI: https://doi.org/10.7554/eLife.41930.004

**Figure supplement 2.** Vpu-mediated counteraction of tetherin.

DOI: https://doi.org/10.7554/eLife.41930.005

Principal component analyses revealed four major clusters, with the donor background accounting for the highest variation in gene expression (*Figure 2B*). Within each donor, the identity of the viral isolate (i.e. CH293, CH077, and STCO1) had more pronounced effects on the cellular transcriptome than the *vpu* genotype (i.e. wt, *vpu* stop, R/K, AA/LL). A similar pattern was observed in a heat map of sample-to-sample distances (*Figure 2—figure supplement 1D*). Differences between the transcriptomes of wt CH293-, CH077- and STCO1-infected cells may reflect specific effects of these viruses on host gene expression and/or stem from differences in overall infection rates. In agreement with the latter hypothesis, higher infection rates (*Figure 2—figure supplement 1E*) resulted in a higher number of differentially expressed genes. For example, HIV-1 CH077 (infecting 60–80% of all CD4 +T cells) significantly modulated the expression of 4202 genes, while CH293 (infecting 20–40% of the cells) resulted in only 1565 differentially expressed genes and STCO1 (infecting 30–60% of the cells) had an intermediate phenotype (*Figure 2—figure supplement 1F*). Importantly, infectious viral doses of the mutants were adjusted to the respective wt viruses for unbiased analysis of the effect of specific Vpu functions on host gene expression (*Figure 2—figure supplement 1E*).

Combined analysis of all three viral isolates revealed that 3369 genes were differentially expressed comparing wt to mock infected CD4 +T cells (1587 up- and 1782 down-regulated) (*Figure 2C*). Gene set enrichment analyses (KEGG) demonstrated that HIV-1 suppresses pathways involved in transcription, translation, cell division and energy metabolism (*Figure 2D*). In agreement with previous analyses in infected T cell lines (*Chang et al., 2011*; *Kleinman et al., 2014*), 'spliceosome', 'RNA transport' and 'ribosome biogenesis' were among the most strongly inhibited pathways. In contrast, gene sets involved in (antiviral) immunity including 'chemokine signaling', 'T cell receptor signaling' and 'antigen processing and presentation' were induced upon wt HIV-1 infection (*Figure 2D*).

These results demonstrate that the three HIV-1 primary isolates CH077, CH293 and STCO1 induce distinct transcription profiles in primary CD4 +T cells that allow us to assess the role of Vpu in immune activation.

## Vpu suppresses NF-κB-dependent antiviral gene expression

To determine the contribution of Vpu to the effects of HIV-1 on host gene expression, we compared the transcriptomes of mutant vs. wt infected cells. Upon complete loss of Vpu (*vpu* stop vs. wt), 92 genes were differentially expressed (54 up, 38 down) across the three HIV-1 clones and four donors tested, while seven genes (six up, one down) were differentially expressed upon selective loss of Vpu-mediated NF-κB inhibition (NF-κB-def. Vpu mutant vs. wt) (*Figure 3A* and *Figure 3—figure supplement 1A*). Notably, all six genes that were upregulated upon introduction of the R/K mutation (i.e. GNG4, NFKBIA, RGS1, NFE2L3, SDC4, NFKB2) were also induced upon complete loss of Vpu. In comparison, abrogation of tetherin counteraction did not significantly affect the expression of any of the genes analyzed (tetherin-def. Vpu mutant vs. wt) (*Figure 3—figure supplement 1A*).

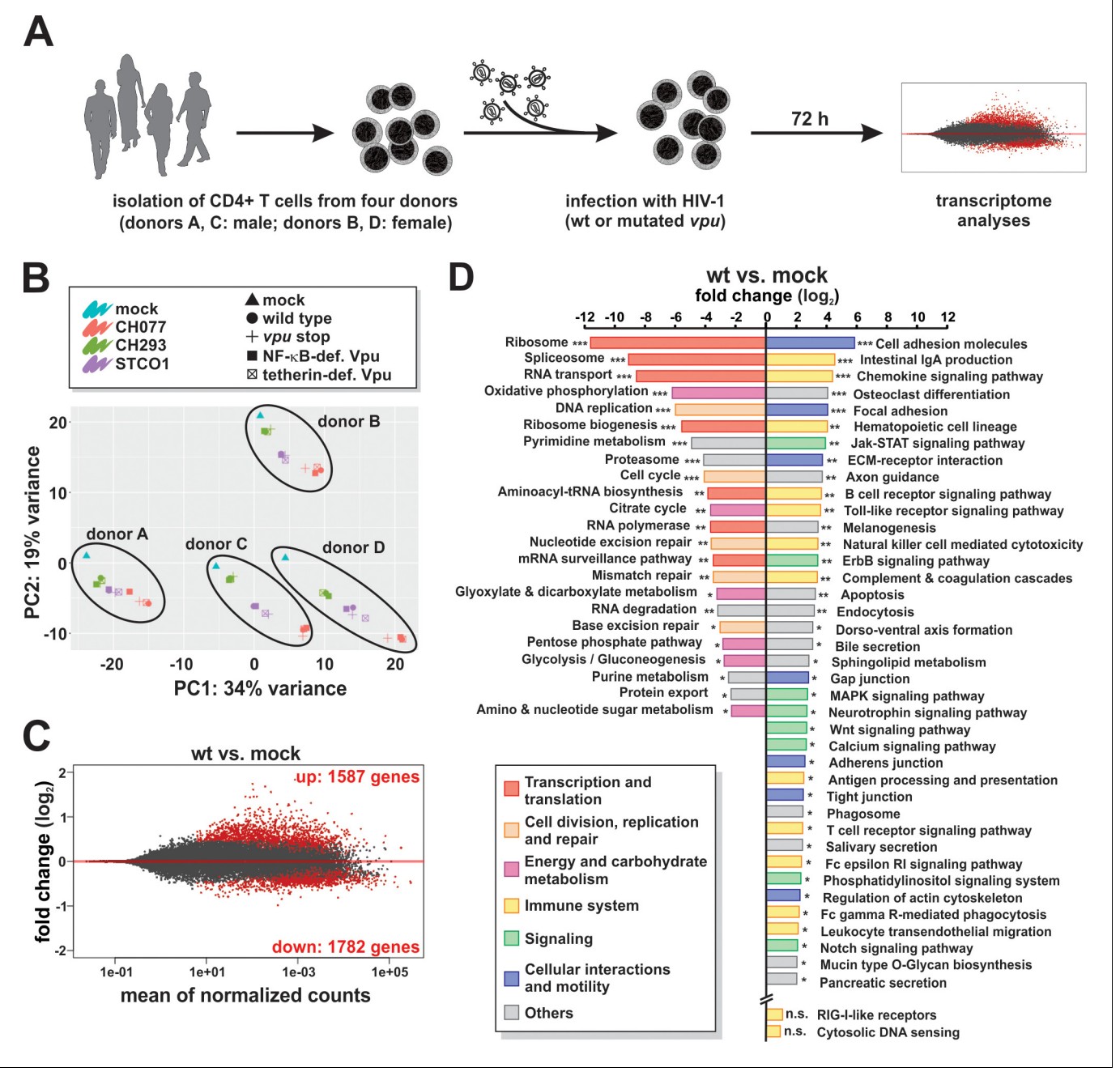

**Figure 2.** RNA sequencing of CD4 +T cells infected with wt or *vpu*-mutated HIV-1. (**A**) Experimental setup. CD4 +T cells of four healthy donors were infected with wt or *vpu*-mutated HIV-1. Three days post-infection, cells were harvested and RNA sequencing was performed. (**B**) Principal component analysis demonstrating that samples are clustered by donor (A - D) and virus (CH077, CH293, STCO1) rather than the *vpu* genotype. (**C**) MA plot providing a global view of differentially expressed genes between Vpu wt and mock infected samples (combined analysis of CH077, CH293, and STCO1). Red dots represent significant genes with a q value < 0.1 after Benjamini-Hochberg correction. (**D**) KEGG pathways significantly down- or up-regulated upon infection with HIV-1 wt (combined analysis of CH077, CH293, and STCO1) (* q < 0.1; ** q < 0.01; *** q < 0.001). HIV-1 suppresses pathways involved in transcription/translation (red), cell division (orange) and energy metabolism (purple), but induces immune responses (yellow), signaling cascades (green) and pathways involved in cell-to-cell interactions and motility (blue). Pathway analysis was performed using GAGE (*Luo et al., 2009*).

DOI: https://doi.org/10.7554/eLife.41930.006

The following figure supplement is available for figure 2:

**Figure supplement 1.** Isolation, infection and RNA sequencing of CD4 +T cells.
DOI: https://doi.org/10.7554/eLife.41930.007

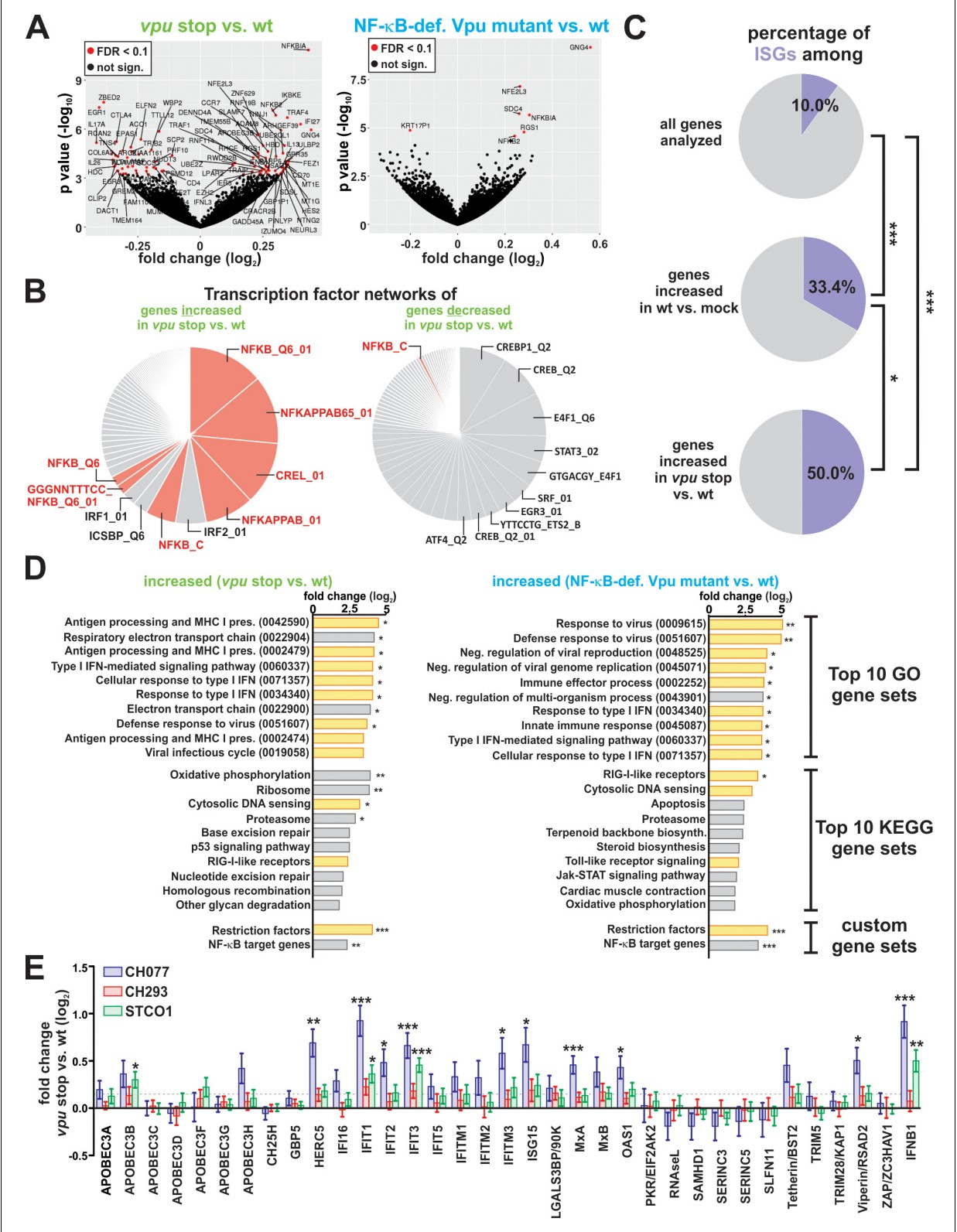

**Figure 3.** Vpu suppresses the induction of NF-κB-dependent immune responses. (**A**) Volcano plots illustrating differentially expressed genes (red) in *vpu* stop vs. wt (left panel) or NF-κB-def. Vpu mutant vs. wt (right panel) infected CD4 +T cells. (**B**) Pie charts illustrating the relative size of gene sets that are targeted by a specific transcription factor. Genes whose expression is significantly increased or decreased upon loss of *vpu* were analyzed in the left and right panel, respectively. NF-κB target gene sets are highlighted in pink. (**C**) Pie charts illustrating the percentage of ISGs among different

*Figure 3 continued on next page*

*Figure 3 continued*
gene sets. The analysis is based on a list of ISGs obtained through the interferome v2.1 database (*Rusinova et al., 2013*) (***p<0.001; *p<0.05; Fisher's exact test). (D) Top10 KEGG and GO pathways that are induced upon complete loss of Vpu (left panel) or selective loss of Vpu-mediated NF-κB inhibition (right panel). Modulation of two custom-defined gene sets (i.e. Restriction factors and NF-κB target genes) is shown at the bottom. Pathways involved in intrinsic, innate or adaptive immunity are highlighted in yellow. Pathway analysis was performed using GAGE (*Luo et al., 2009*) (* q < 0.1; ** q < 0.01; *** q < 0.001). (E) n-fold modulation of host restriction factor and IFN-β mRNA levels upon loss of Vpu. Mean values of four independent infections ± SEM are shown. Statistical significance was determined using negative binomial generalized linear models as implemented in DESeq2 (*Love et al., 2014*) (* q < 0.1; ** q < 0.01; *** q < 0.001). The dotted line indicates the mean change of restriction factor expression upon loss of Vpu.
DOI: https://doi.org/10.7554/eLife.41930.008
The following figure supplement is available for figure 3:

**Figure supplement 1.** Modulation of NF-κB-dependent gene expression by Vpu.
DOI: https://doi.org/10.7554/eLife.41930.009

This agrees with our result that the AA/LL mutants is still able to prevent NF-κB activation downstream of tetherin (*Figure 1A*).

To determine whether the 92 genes modulated by Vpu share a common regulatory mechanism, we performed transcription factor network analyses based on the transcription factor target (TFT) gene sets of the Molecular Signatures Database (MSigDB) (*Xie et al., 2005*). This unbiased approach revealed that Vpu preferentially suppresses the expression of NF-κB target genes, with gene sets regulated by other transcription factors (including IRFs) playing only a minor role (*Figure 3B*, left panel). In contrast, genes upregulated by Vpu (*Figure 3B*, right panel) showed no enrichment of NF-κB target genes. The latter observation also excludes a potential bias due to the high number of NF-κB target genes in the human genome.

In total, several hundred genes can be upregulated upon NF-κB activation, either directly or indirectly. Of the more than 26.000 cellular transcripts covered by our RNA-Seq approach, about 3% are NF-κB inducible (*Figure 3—figure supplement 1B*). This percentage increases to 30% and 67% in genes, whose expression is up-regulated upon complete loss of Vpu or selective loss of Vpu-mediated NF-κB inhibition, respectively (*Figure 3—figure supplement 1B and C*). In contrast, we did not find any of 34 previously identified IRF3 targets (*Grandvaux et al., 2002*) among the transcripts decreased in the presence of Vpu (data not shown).

In agreement with a key role of NF-κB in antiviral immunity, IFN-stimulated genes were also enriched among genes suppressed by Vpu (*Figure 3C*). Among them were signaling molecules such as TRAF1, TRAF4, or IKKε, but also previously described Vpu targets including the NK cell activating receptor ULBP2 and the chemokine receptor CCR7 (*Figure 3—figure supplement 1D*, see arrows) (*Galaski et al., 2016*; *Ramirez et al., 2014*). GO and KEGG gene set enrichment analyses demonstrated that Vpu preferentially inhibits the induction of pathways involved in innate and adaptive immunity including 'response to type I IFN' and 'antigen processing and MHC I presentation' (*Figure 3D*, left panel). Intriguingly, 'cytosolic DNA sensing' and 'RIG-I like receptors' were not significantly induced by wt HIV-1 infection, but markedly upregulated upon disruption of *vpu* (*Figures 2D* and *3D* left panel). A similar phenotype was observed upon selective abrogation of Vpu-mediated NF-κB inhibition (*Figure 3D*, right panel), strongly suggesting that these immunosuppressive effects of Vpu depend on its ability to prevent NF-κB activation.

As several host restriction factors (e.g. IFIT1, ISG15, HERC5) are expressed in an NF-κB-dependent manner (*Kroismayr et al., 2004*; *Pfeffer et al., 2004*), we defined two additional custom gene sets: 'NF-κB target genes' (*Supplementary file 2*; http://bioinfo.lifl.fr/NF-KB/) and 'Host restriction factors' (*Supplementary file 3*; *Kluge et al., 2015*). Again, both of them were significantly upregulated upon loss of Vpu or Vpu-mediated NF-κB inhibition (*Figure 3D*, bottom). Analysis of individual genes revealed that Vpu significantly reduced mRNA levels of the host restriction factors APOBEC3B, HERC5, IFIT1-3, IFITM3, ISG15, MxA, OAS1 and Viperin, as well as IFN-β (*Figure 3E*). In summary, these findings demonstrate that HIV-1 Vpu exerts broad immune-suppressive activity in infected primary CD4 +T cells.

## Vpu suppresses pro-inflammatory cytokine production

We selected pro-inflammatory cytokines such as IFN-β, IL-6, CXCL10/IP-10 for validation of our RNA-Seq results. These factors represent known NF-κB targets with key roles in anti-viral immunity

(*Libermann and Baltimore, 1990*; *Ohmori and Hamilton, 1993*), disease progression of HIV-1 infected individuals (*Hamlyn et al., 2014*; *Jiao et al., 2012*; *Noel et al., 2014*), and/or reactivation of latent viral reservoirs (*Hoshino et al., 2010*). Quantitative RT-PCR confirmed that lack of Vpu-dependent inhibition of NF-κB activation was associated with 2- to 11-fold increased IFN-β mRNA expression levels in HIV-1 infected CD4 +T cells (*Figure 4A*, top panels). In case of HIV-1 CH077, the R45K mutation had only modest effects, which is in agreement with its incomplete loss of NF-κB inhibition (*Figure 1C*). Overall, the effects of the viruses on IFN-β induction correlated well with their modulation of NF-κB activity (*Figure 4A*, bottom panels). Importantly, differences in IFN-β expression were not a consequence of different infection rates (*Figure 4—figure supplement 1A*).

To validate our RNA-Seq results at the protein level, we used a flow-cytometry-based approach to quantify cytokine concentrations in the culture supernatants of CD4 +T cells at day three post-infection. We selected STCO1 for our analyses as this strain results in high infection rates in CD4 +T cells and shows a complete loss of NF-κB inhibition upon mutation of R45K. As before, virus stocks were normalized for wt and mutant viruses (*Figure 4—figure supplement 1B*). The AA/LL mutations abrogating tetherin counteraction had no marked effect on CXCL10 or IL-6 production. In contrast, loss of Vpu-mediated NF-κB inhibition increased expression of these pro-inflammatory cytokines by 50% to 300% (*Figure 4B*). This result was confirmed in a membrane array-based approach that allows simultaneous detection of 80 different cytokines. In addition to CXCL10 and IL-6, Vpu suppressed the production of MIP-1β/CCL4, RANTES/CCL5, TNF-α/-β, IFN-γ and many additional cytokines in an NF-κB-dependent manner (*Figure 4C*). On average, complete lack of Vpu resulted in 32% and the single R45K mutation in 24% increases of NF-κB-inducible cytokine levels, respectively. Thus, qRT-PCR and protein arrays confirmed the RNA-Seq results, demonstrating that Vpu suppresses pro-inflammatory cytokine expression via its ability to inhibit NF-κB activation.

## Discussion

Besides its ability to decrease CD4 protein levels at the cell surface and to counteract tetherin, accumulating evidence suggested that HIV-1 Vpu exerts much broader immunosuppressive effects by inhibiting the activation of cellular transcription factors (*Sauter and Kirchhoff, 2018*). In the present study, we therefore elucidated (1) the global impact of Vpu on host gene expression, (2) the transcription factors that are targeted by Vpu, and (3) the role of tetherin counteraction in the immunosuppressive activity of Vpu. Combining the generation of selective *vpu* mutants of primary HIV-1 clones with a broad and unbiased RNA sequencing approach of infected CD4 +T cells, we show that Vpu suppresses the transcription of a plethora of NF-κB-inducible host genes with key roles in intrinsic, innate and adaptive immune responses. qPCR analyses and quantification of cytokine release by two independent methods confirmed that Vpu reduces the expression of type I IFNs and other pro-inflammatory cytokines (*Figure 4*).

Since immune-modulatory functions are frequently impaired in lab-adapted HIV-1 strains (*Apps et al., 2016*; *Pickering et al., 2014*), we exclusively used clones representing primary HIV-1 isolates, including one transmitted/founder virus. The latter establish successful infection upon transmission and are characterized by an increased resistance to type I IFNs that enables efficient replication in an environment of augmented immune activation observed during acute infection (*Iyer et al., 2017*; *Parrish et al., 2013*).

Selective abrogation of tetherin antagonism did not significantly change host gene expression, demonstrating that tetherin-independent inhibition of NF-κB activation is sufficient for the immunosuppressive activity of Vpu. In agreement with this, a single arginine-to-lysine mutation abrogating the ability of Vpu to inhibit NF-κB activation downstream of tetherin also resulted in significantly increased expression of NF-κB target genes and the induction of various immunity pathways. Nevertheless, introduction of the R/K mutation had much less pronounced effects on host gene expression than complete disruption of Vpu (*Figure 3—figure supplement 1A,C,D*). These differences may be explained by NF-κB-independent effects of Vpu on transcription and/or result from an only partial loss of NF-κB inhibition in the R/K mutants (*Figure 1C*).

The exact pathways triggering NF-κB activation in HIV-1 infected CD4 +T cells remain to be determined. Although a number of innate HIV sensors (e.g. IFI16, cGAS, PQBP1) have been shown to efficiently induce the activation of IRF3 and/or IRF7 (*Jakobsen et al., 2013*; *Krapp et al., 2018*; *Yoh et al., 2015*), their relative contribution to NF-κB activation is still unclear. Besides pattern

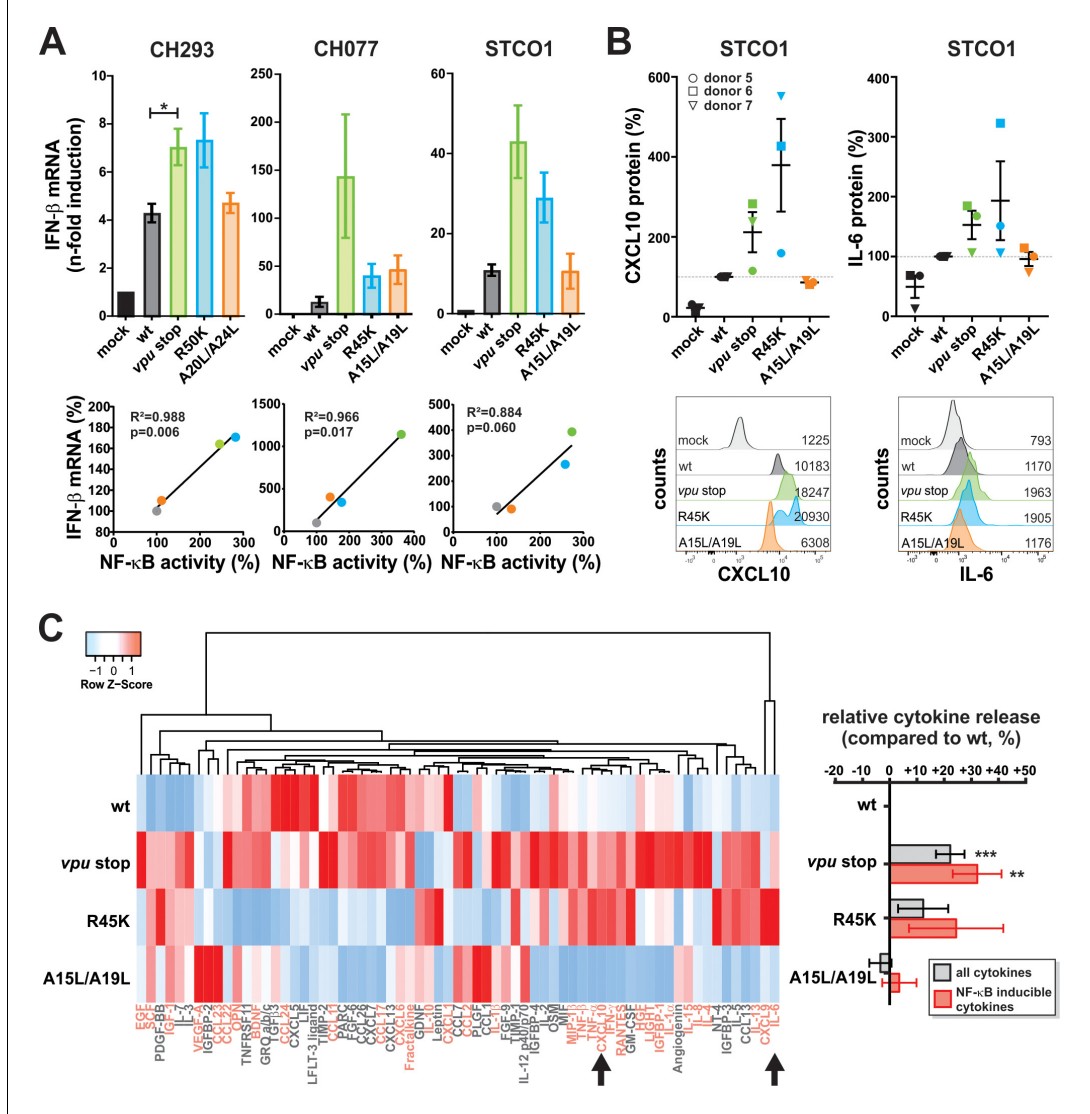

**Figure 4.** Vpu suppresses pro-inflammatory cytokine production. (**A**) Vpu inhibits IFN-β expression. CD4 +T cells were infected with the indicated VSV-G pseudotyped HIV-1 strains. Three days post-infection, IFN-β mRNA levels were quantified by qRT-PCR and normalized to the respective mock control. In the top panels, mean values of three to four independent experiments ± SEM are shown. The bottom panels show correlation analyses of IFN-β mRNA levels and NF-κB activity (determined in **Figure 1C**) (*p<0.05; RM one-way ANOVA with Greenhouse-Geisser correction and Dunnett's multiple comparison test). (**B**) Vpu suppresses release of CXCL10 and IL-6. CD4 +T cells were infected with the indicated VSV-G pseudotyped HIV-1 STCO1 mutants. Three days post-infection, cytokine concentrations in the cell culture supernatant were quantified by flow cytometry using a bead-based immunoassay, in which the analytes are captured between two antibodies. In the top panels, each data point represents results obtained with one of three donors, normalized to the wild type control (100%). Error bars indicate SEM. Representative primary data of one donor are shown in the bottom panels. Numbers indicate mean fluorescence intensity. (**C**) Vpu exerts broad suppressive effects on cytokine release. CD4 +T cells were infected with the indicated VSV-G pseudotyped HIV-1 STCO1 mutants. Three days post-infection, cytokine release was determined using a cytokine array, based on antibody-coated membranes. The heat map on the left illustrates relative cytokine release and is based on the mean of three independent experiments. Cytokines that were not detectable in more than one donor were omitted from the heat map. Known NF-κB inducible genes (70) are shown in pink. Arrows indicate CXCL10 and IL-6 that were also analyzed in panel (**B**). Mean changes ± SEM in cytokine release relative to HIV-1 wt are shown in the right panel. (**p<0.01; ***p<0.001; RM one-way ANOVA with Greenhouse-Geisser correction and Dunnett's multiple comparison test).
DOI: https://doi.org/10.7554/eLife.41930.010
The following figure supplement is available for figure 4:

**Figure supplement 1.** Infection rates of CD4 +T cells.
DOI: https://doi.org/10.7554/eLife.41930.011

recognition receptors, activation via the T cell receptor complex may be an important determinant of NF-κB activation (*Jamieson et al., 1991*). While our unbiased approach supports the hypothesis that Vpu preferentially targets NF-κB-driven gene expression, it does not confirm previous studies suggesting that Vpu suppresses the expression of IRF-3 target genes (*Doehle et al., 2012*; *Park et al., 2014*). Interestingly, however, some of the genes targeted by Vpu share a regulation by IRF-1, IRF-2 and/or IRF-8/ICSBP (*Figure 3B*). These transcription factors are IFN-γ-inducible and may cooperate with each other to prevent prolonged expression of IFN-stimulated genes (*Elser et al., 2002*; *Horiuchi et al., 2011*; *Kuwata et al., 2002*; *Tamura et al., 2005*). However, the observed effect of Vpu on IRF-1,–2 and −8 and their target genes may only be indirect since Vpu reduced the release of IFN-γ, an NF-κB target gene, from infected cells (*Figure 4C*).

Besides IFN-γ, Vpu also suppressed the production of other cytokines, including IFN-β, CXCL10/IP-10, IL-6 and TNF-α (*Figure 4*), even in the context of PHA-prestimulated CD4 +T cells. All these factors are part of the cytokine storm during acute lentiviral infection (*Abel et al., 2005*; *Huang et al., 2016*; *Stacey et al., 2009*), but fail to eliminate the virus. These pro-inflammatory cytokines may even contribute to disease progression and accelerate pathogenesis of AIDS by inducing prolonged hyper-activation of the immune system. For example, increased plasma levels of CXCL10, TNF-α and IL-6 are associated with accelerated loss of CD4 +T cells in HIV-1 infected individuals (*Borges et al., 2015*; *Liovat et al., 2012*; *Vaidya et al., 2014*). During primary HIV-1 infection, CXCL10 plasma levels predict disease progression even better than plasma viral loads or CD4 +T cell counts (*Liovat et al., 2012*). By modulating cellular cytokine production, Vpu may therefore be an important determinant of HIV-1 pathogenesis. Notably, however, the suppression of immune activation by Vpu is incomplete, and wt HIV-1 still triggers the activation of a variety of immune signaling pathways, including T cell, B cell, Fc and Toll-like receptor signaling (*Figure 2D*). In fact, 'RIG-I-like receptors' and 'cytosolic DNA sensing' were the only immunity-related KEGG pathways that were not significantly upregulated upon wt HIV-1 infection (*Figure 2D*). Thus, incomplete suppression of NF-κB activation and/or the induction of pathways involving transcription factors that are not directly targeted by Vpu may underlie the detrimental immune activation that drives disease progression in HIV-1 infected individuals.

We also found that several host restriction factors (i.e. APOBEC3B, HERC5, IFIT1-3, IFITM3, ISG15, MxA, OAS1 and Viperin) are significantly upregulated upon loss of Vpu (*Figure 3E*). The promoters of many restriction factors contain both NF-κB- and IRF-responsive elements (*Pfeffer et al., 2004*). Thus, this increase may either be a direct consequence of missing Vpu-mediated NF-κB inhibition and/or the result of elevated cytokine levels. Our data suggest that HIV-1 uses Vpu to evade certain restriction factors by counteracting them at the transcriptional level. In agreement with this, we have previously shown that Vpu not only targets tetherin at the protein level, but also reduces its mRNA expression (*Sauter et al., 2015*). Although many restriction factors are constitutively expressed, their expression and hence antiviral effect usually increases upon infection and/or IFN simulation. Thus, Vpu-mediated suppression of transcription may considerably relieve restriction. In this regard, it is interesting to note, that no direct HIV-1 antagonist has been identified for most restriction factors suppressed by Vpu (e.g. IFITs, HERC5, ISG15).

Among the differentially expressed genes, we also found several host factors previously described as targets of Vpu (*Figure 3—figure supplement 1D*). For example, flow cytometric analyses of HIV-1 wt- and Δ*vpu*-infected cells demonstrated that Vpu decreases cell surface levels of the NK cell activating ligands ULBP2/5/6 (*Galaski et al., 2016*) and the chemokine receptor CCR7 (*Ramirez et al., 2014*). Our RNA sequencing results suggest that Vpu does not (or at least not exclusively) target these factors at the protein level but also prevents their expression by inhibiting the activation of NF-κB.

Our results clearly demonstrate a broad immunosuppressive activity of Vpu in ex vivo infected primary CD4 +T cells. Furthermore, they are in agreement with the model that HIV-1 Nef boosts NF-κB activation during early stages of infection to initiate efficient transcription of viral genes, while Vpu suppresses NF-κB activation during later stages to limit antiviral gene expression of the host (*Sauter et al., 2015*). However, the importance of this Vpu activity for viral replication, immune activation and disease progression during different stages of infection remains to be determined in vivo. One limitation of the present study is the use of VSV-G for pseudotyping, which may influence sensing and overall gene expression. Furthermore, the use of unsorted cells does not allow to discriminate between effects in infected and uninfected cells and may introduce a bias with respect to

infection rates. Nevertheless, this potential confounder does not affect the comparison of wild type and *vpu* mutated viruses as the variants of each strain were normalized for their infectivity to obtain similar infection rates (*Figure 2—figure supplement 1E*). In fact, the present approach may even underestimate the inhibitory effects of Vpu due to the presence of 25–80% uninfected bystander cells and the pre-activation of CD4 +T cells with PHA.

It also remains to be clarified whether the inhibition of NF-κB activation by Vpu may compensate for the loss of Nef-mediated down-modulation of the T cell receptor CD3 in HIV-1 and its direct simian precursors. The selective Vpu mutants generated in this study provide a valuable means to address these questions in various in vivo models, including humanized mice (*Yamada et al., 2015*) and a recently established monkey model (*Joas et al., 2018*). An important role of Vpu-mediated NF-κB inhibition for efficient viral spread and replication in vivo is supported by the observation that this activity is highly conserved among HIV-1 and closely related simian immunodeficiency viruses (SIV) (*Sauter et al., 2015*). In contrast, HIV-2 and the majority of SIV species, which do not encode Vpu, have evolved alternative mechanisms and prevent the activation of NF-κB via their accessory proteins Nef or Vpr (*Hotter et al., 2017*). In fact, components of the canonical NF-κB signaling cascade are targeted by diverse virus families. Well-described examples are IκB-like proteins encoded by polydnaviruses (*Falabella et al., 2007*), the Vaccinia virus K1 protein, which prevents the acetylation of RelA (*Bravo Cruz and Shisler, 2016*) or African swine fever virus A238L, which binds p65/NF-κB to prevent its nuclear translocation (*Revilla et al., 1998*). These examples of convergent viral evolution highlight the importance of NF-κB in the antiviral immune response and demonstrate that Vpu-mediated NF-κB inhibition is one of several sophisticated mechanisms that viruses have evolved to prevent the manifold immune responses triggered by this transcription factor.

# Materials and methods

**Key resources table**

| Reagent type (species) or resource | Designation | Source or reference | Identifiers | Additional information |
|---|---|---|---|---|
| Gene (*H. sapiens*) | Custom-defined gene set of NF-κB target genes | see *Supplementary file 2* of this paper | N/A | |
| Gene (*H. sapiens*) | Custom-defined gene set of host restriction factors | see *Supplementary file 3* of this paper | N/A | |
| Strain, strain background (*E. coli*) | XL-2 blue | Stratagene | Cat#200150 | |
| Strain, strain background (*E. coli*) | XL2-Blue MRF' TM Ultracompetent cells | Agilent Technologies | Cat#200151 | |
| Cell line (*H. sapiens*) | HEK293T, female | ATCC | Cat#CRL-3216; RRID: CVCL_0063 | |
| Cell line (*H. sapiens*) | TZM-bl, female | NIH | Cat#8129; RRID: CVCL_B478 | |
| Biological sample (*H. sapiens*) | peripheral blood mononuclear cells (donors A and C: male; donors B and D: female) | DRK-Blutspendedienst Baden-Württemberg-Hessen, Ulm, Germany | N/A | derived from four healthy donors |
| Antibody | anti-HIV-1 p24 (capsid protein), FITC-conjugated (mouse monoclonal) | Beckman Coulter | Cat#6604665; RRID: AB_1575987 | FACS (1:25) |
| Antibody | anti-BST-2/tetherin, APC-conjugated (mouse monoclonal) | Biolegend | Cat#348410; RRID:AB_2067121 | FACS (1:20) |

*Continued on next page*

Continued

| Reagent type (species) or resource | Designation | Source or reference | Identifiers | Additional information |
|---|---|---|---|---|
| Antibody | anti-Human CD4, PerCP-conjugated (mouse monoclonal) | BD Pharmingen | Cat#550631; RRID:AB_393791 | FACS (1:12.5) |
| Antibody | anti-CD11c [3.9], FITC-conjugated (mouse monoclonal) | Abcam | Cat#ab82445-100; RRID:AB_1859733 | FACS (1:12.5) |
| Antibody | anti-CD11c [BU15], FITC-conjugated (mouse monoclonal) | Abcam | Cat#ab22540 | FACS (1:5) |
| Antibody | IgG1, κ, APC-conjugated (mouse) | Biolegend | Cat#400122; RRID:AB_326443 | FACS (1:20) |
| Antibody | HIV-1 subtype C serum | NIH | Cat#11942 | WB (1:5000) |
| Antibody | HIV-1 NL4-3 Vpu antiserum | N/A | N/A | WB (1:500); kindly provided by S. Bolduan |
| Antibody | Anti-HIV1 p24 antibody [39/5.4A] | Abcam | Cat#ab9071; RRID:AB_306981 | WB (1:5000) |
| Antibody | Purified anti-GAPDH antibody | Biolegend | Cat#607902; RRID:AB_2734503 | WB (1:1000) |
| Antibody | IRDye 800CW Goat anti-Mouse IgG (H + L) | LI-COR | Cat#926–32210; RRID:AB_621842 | WB (1:20000) |
| Antibody | IRDye 680RD Goat anti-Rabbit IgG (H + L) | LI-COR | Cat#925–68071; RRID:AB_2721181 | WB (1:20000) |
| Antibody | IRDye 800CW Goat anti-Rat IgG (H + L) | LI-COR | Cat#925–32219; RRID:AB_2721932 | WB (1:20000) |
| Antibody | anti-HIV-1 p24 (mouse monoclonal) | Abcam | Cat#ab9071; RRID:AB_306981 | IF (1:500) |
| Antibody | anti-p65 (rabbit polyclonal IgG) | Santa Cruz | Cat#sc-372; RRID:AB_632037 | IF (1:150) |
| Antibody | Donkey anti-Mouse IgG (H + L), AF647-conjugated | Thermo Scientific | Cat#A31571; RRID:AB_162542 | IF (1:350) |
| Antibody | Goat anti-Rabbit IgG (H + L), AF568-conjugated | Thermo Scientific | Cat#A11011; RRID:AB_143157 | IF (1:350) |
| Recombinant DNA reagent | pBR322_HIV-1 M subtype B STCO1 wt (plasmid) | PMID: 23542380 | N/A | kindly provided by B. Hahn |
| Recombinant DNA reagent | pBR322_HIV-1 M subtype B STCO1 vpu stop (plasmid) | PMID: 27531907 | N/A | |
| Recombinant DNA reagent | pBR322_HIV-1 M subtype B STCO1 Vpu R45K (plasmid) | PMID: 29324226 | N/A | |
| Recombinant DNA reagent | pBR322_HIV-1 M subtype B STCO1 Vpu A15L/A19L (plasmid) | PMID: 27531907 | N/A | |
| Recombinant DNA reagent | pUC57rev_HIV-1 M subtype C CH293 w8 wt (plasmid) | PMID: 23542380 | N/A | kindly provided by B. Hahn |

*Continued*

| Reagent type (species) or resource | Designation | Source or reference | Identifiers | Additional information |
|---|---|---|---|---|
| Recombinant DNA reagent | pUC57rev_HIV-1 M subtype C CH293 w8 vpu stop (plasmid) | PMID: 2562070 | N/A | |
| Recombinant DNA reagent | pUC57rev_HIV-1 M subtype C CH293 w8 Vpu R50K (plasmid) | this paper | N/A | derived from pUC57 rev_HIV-1 M subtype C CH293 w8 wt (plasmid) |
| Recombinant DNA reagent | pUC57rev_HIV-1 M subtype C CH293 w8 Vpu A20L /A24L (plasmid) | this paper | N/A | derived from pUC57 rev_HIV-1 M subtype C CH293 w8 wt (plasmid) |
| Recombinant DNA reagent | pCR-XL-TOPO_HIV-1 M subtype B CH077 wt (plasmid) | PMID: 22190722 | N/A | kindly provided by B. Hahn |
| Recombinant DNA reagent | pCR-XL-TOPO_HIV-1 M subtype B CH077 vpu stop (plasmid) | PMID: 27531907 | N/A | |
| Recombinant DNA reagent | pCR-XL-TOPO_HIV-1 M subtype B CH077 Vpu R45K (plasmid) | this paper | N/A | derived from pCR-XL-TOPO_HIV-1 M subtype B CH077 wt (plasmid) |
| Recombinant DNA reagent | pCR-XL-TOPO_HIV-1 M subtype B CH077 Vpu A15L/A19L (plasmid) | PMID: 27531907 | N/A | |
| Recombinant DNA reagent | p_human IKKβ, constitutively active mutant (S177E, S181E) (plasmid) | PMID: 23552418 | N/A | kindly provided by B. Baumann |
| Recombinant DNA reagent | p_NF-κB(3x)-Firefly Luciferase (plasmid) | PMID: 23552418 | N/A | kindly provided by B. Baumann |
| Recombinant DNA reagent | pTAL_*Gaussia* Luciferase (plasmid) | PMID: 23552418 | N/A | |
| Recombinant DNA reagent | pCG_human Tetherin IRES DsRed2 (plasmid) | PMID: 19917496 | N/A | |
| Recombinant DNA reagent | pHIT-G_VSV-G (vesicular stomatitis virus glycoprotein) (plasmid) | PMID: 9303297 | N/A | |
| Sequence-based reagent | Primers used for mutagenesis of vpu | see *Supplementary file 4* of this paper | N/A | |
| Sequence-based reagent | TaqMan Gene Expression Assay for IFN-beta | Thermo Fisher Scientific | Cat#Hs01077958_s1 | |
| Sequence-based reagent | Human GAPD (GAPDH) Endogenous Control (VIC/TAMRA probe, primer limited) | Thermo Fisher Scientific | Cat#4310884E | |
| Peptide, recombinant protein | IRDye 800CW streptavidin, 0.5 mg | LI-COR | Cat#926–32230 | |
| Commercial assay or kit | GalScreen | Applied Bioscience | Cat#T1027 | |
| Commercial assay or kit | PrimeScript RT-PCR Kit | TAKARA | Cat#RR014A | |

*Continued on next page*

*Continued*

| Reagent type (species) or resource | Designation | Source or reference | Identifiers | Additional information |
|---|---|---|---|---|
| Commercial assay or kit | RNeasy Plus Mini kit | QIAGEN | Cat#74136 | |
| Commercial assay or kit | RosetteSep Human CD4 + T Cell Enrichment Cocktail | Stem Cell Technologies | Cat#15062 | |
| Commercial assay or kit | Luciferase Assay System 10-pack | Promega | Cat#E1501 | |
| Commercial assay or kit | LEGENDplex Human Anti-Virus Response Panel | BioLegend | Cat#740390 | |
| Commercial assay or kit | RayBio C-Series Human Cytokine Antibody Array C5 | RayBiotech | Cat#AAH-CYT-5–8 | |
| Commercial assay or kit | FIX and PERM Kit (CE-IVD) (1000 Tests) | Nordic-MUbio | Cat#GAS-002–1 | |
| Commercial assay or kit | DNA-free DNA Removal Kit | ThermoFisher Scientific | Cat#AM1906 | |
| Commercial assay or kit | QuikChange II XL Site-Directed Mutagenesis Kit | Agilent | Cat#200522 | |
| Commercial assay or kit | TruSeq Stranded mRNA Sample Prep Kit | illumina | Cat#RS-122–2101 | |
| Commercial assay or kit | Thermo Script RT-PCR system | Invitrogen | Cat#11146–016 | |
| Commercial assay or kit | Signal Enhancer HIKARI kit for Western Blotting and ELISA | Nacalai Tesque | Cat#02267–41 | |
| Chemical compound, drug | Human IL-2 IS, premium grade | MACS Miltenyi Biotec | Cat#130-097-745 | |
| Chemical compound, drug | Remel PHA purified | ThermoFisher Scientific | Cat#R30852801 | |
| Chemical compound, drug | Recombinant human TNFα | Sigma Aldrich | Cat#H8916-10UG | |
| Chemical compound, drug | DAPI | Sigma Aldrich | Cat#D9542-1MG | |
| Software, algorithm | BD FACSDiva Version 8.0 | BD Biosciences | https://www.bdbiosciences.com; RRID: SCR_001456 | |
| Software, algorithm | Corel DRAW 2017 | Corel Corporation | https://www.coreldraw.com/ | |
| Software, algorithm | GraphPad Prism Version 5.03 | GraphPad Software, Inc. | https://www.graphpad.com; RRID: SCR_002798 | |
| Software, algorithm | ImageJ | Open source | http://imagej.nih.gov/ij/ | |
| Software, algorithm | LI-COR Image Studio Lite Version 3.1 | LI-COR | www.licor.com/; RRID: SCR_013715 | |
| Software, algorithm | FlowJo_V10 | Tree Star, Inc. | https://www.flowjo.com | |

*Continued on next page*

*Continued*

| Reagent type (species) or resource | Designation | Source or reference | Identifiers | Additional information |
|---|---|---|---|---|
| Software, algorithm | ZEN | Zeiss | https://www.zeiss.com/microscopy/int/products/microscope-software/zen.html; RRID:SCR_013672 | |
| Software, algorithm | Fiji | Open source (Max Planck Institute of Molecular Cell Biology and genetics, Dresden, Germany) | https://imagej.net/Fiji; RRID:SCR_002285 | |
| Software, algorithm | GeneMANIA | Open source | http://genemania.org/ | |
| Software, algorithm | Heatmapper | Open source | http://www2.heatmapper.ca/expression/ | |
| Software, algorithm | R | Open source | https://www.r-project.org/ | |
| Software, algorithm | Kallisto | PMID: 27043002 | https://pachterlab.github.io/kallisto/ | |
| Software, algorithm | DESeq2 | PMID: 25516281 | https://bioconductor.org/packages/release/bioc/html/DESeq2.html | |
| Software, algorithm | GAGE | PMID: 19473525 | https://bioconductor.org/packages/release/bioc/html/gage.html | |
| Software, algorithm | StepOne and Step OnePlus Software v2.3 | Thermo Fisher Scientific | https://www.thermofisher.com/de/de/home/technical-resources/software-downloads/StepOne-and-StepOnePlus-Real-Time-PCR-System.html | |
| Other | Rel/NF-κB target gene database | Open source | http://bioinfo.lifl.fr/NF-KB/ | |
| Other | RNAseq Data | this paper | GEO accession GSE117655 | |
| Other | Interferome database | PMID: 23203888 | http://www.interferome.org/interferome/home.jspx | |

## Cell culture

Human embryonic kidney 293T (HEK293T, RRID: CVCL_0063), obtained from the American Type Culture Collection (ATCC), was first described by DuBridge and colleagues (*DuBridge et al., 1987*). TZM-bl cells (RRID: CVCL_B478) are a HeLa-derived reporter cell line and were obtained through the NIH AIDS Reagent Program, Division of AIDS, NIAID, NIH, from John C Kappes, Xiaoyun Wu, and Tranzyme, Inc (*Platt et al., 1998*). HEK293T and TZM-bl cells were authenticated by the ATCC and NIH, respectively and maintained in Dulbecco's modified Eagle medium (DMEM) supplemented with 10% FCS, 2 mM glutamine, streptomycin (100 µg/ml), penicillin (100 U/ml). Both cell lines were tested for mycoplasma contamination every three months. Only mycoplasma negative cells were used for this study.

PBMCs were isolated using lymphocyte separation medium (Biocoll separating solution; Biochrom, Berlin, Germany). CD4 +T cells were isolated using the RosetteSep Human CD4 +T Cell Enrichment Cocktail (Stem Cell Technologies, Vancouver, Canada). Primary cells were cultured in RPMI-1640 containing 10% fetal calf serum (FCS), 2 mM glutamine, streptomycin (100 μg/ml), penicillin (100 U/ml), and 66 IU/ml = 10 ng/ml interleukin 2 (IL-2). Before infection, primary cells were stimulated for three days with phytohemagglutinin (PHA) (1 μg/ml). The use of human PBMCs was approved by the Ethics Committee of the Ulm University Medical Center. All donors were anonymized and provided informed written consent.

## Proviral constructs and mutagenesis

Infectious molecular clones of HIV-1 CH293, CH077 and STCO1 were kindly provided by Beatrice H. Hahn (*Ochsenbauer et al., 2012*; *Parrish et al., 2013*). *Vpu* stop mutations were introduced using the QuikChange II XL Site-Directed Mutagenesis Kit (Agilent, Santa Clara, CA); all other *vpu* mutations were introduced using SOE-PCR-based mutagenesis (see *Supplementary file 4* for primers).

## Generation of virus stocks

HEK293T cells were co-transfected in a 6-well format with proviral constructs (5 μg) and an expression plasmid for the vesicular stomatitis virus glycoprotein (pHIT_VSV-G; 1 μg) (*Fouchier et al., 1997*) using the calcium phosphate method. Two days post-transfection, cell culture supernatants were harvested and cleared by centrifugation (1700 g, 4 min, 4°C, Centrifuge 5417C (Eppendorf, Hamburg, Germany), fixed-angle rotor F-45-30-11). For infection of primary CD4 +T cells (RNA-Seq, cytokine arrays, qRT-PCR), virus stocks were concentrated 20-fold via ultracentrifugation (96325 g, 120 min, 4°C, UC OptimaTM L-80 XP Ultracentrifuge (Beckman Coulter, Brea, CA), Swinging-Bucket Rotor SW Ti-32) and for each of the three HIV-1 strains, wild type and mutant viruses were adjusted for infectivity using a TZM-bl reporter cell assay.

## Infection of TZM-bl reporter cells

6,000 TZM-bl cells were seeded in 96-well plates and infected with 1–100 μl cell culture supernatant in triplicate on the following day. Three days post-infection, *β-galactosidase* reporter gene expression was determined using the GalScreen Kit (Applied Biosystems, Foster City, CA).

## Infection of PBMCs and CD4 +T cells

To determine Vpu-mediated effects on tetherin surface levels, 1 million activated PBMCs were infected with 300 μl non-concentrated VSV-G-pseudotyped HIV-1 for 6 hr in 4 ml tubes before culturing them in 2 ml RPMI-1640 in 6-wells. CD4 +T cells were infected with concentrated virus stocks via spinoculation (2500 g, 90 min (RNA-Seq) or 1200 g, 120 min (qRT-PCR, cytokine release assays), 37°C, Centrifuge 5810R (Eppendorf, Hamburg, Germany), Swing-bucket rotor A-4–81).

## Flow cytometry

Infection rates and Vpu-mediated tetherin down-modulation in infected primary cells were determined three days post-infection. To this end, cells were stained extracellularly for tetherin (BioLegend, San Diego, CA, Cat# 348410, dilution 1:20) before fixation, permeabilization (FIX and PERM Kit, Nordic-MUbio, Susteren, Netherlands) and staining for HIV-1 p24 (Beckman Coulter, Brea, CA, Cat# 6604665, RRID: AB_1575987, dilution 1:25). Between 1000 and 6000 infected (i.e. p24+) cells were acquired per sample to monitor tetherin surface expression, whereas 14,000 to 450,000 live cells were acquired to determine infection rates in *Figure 2—figure supplement 1E* and *Figure 4—figure supplement 1A,B*. To analyze the purity of CD4 +T cells after isolation, cells were simultaneously stained for surface CD4 (BD Pharmingen, San Jose, CA, Cat#550631, dilution 1:12.5) and CD11c (Abcam, Cambridge, UK, Cat# ab22540, dilution 1:5 or #ab82445-100, RRID: AB_1859733, dilution 1:12.5). In this case, 50,000 to 58,000 live cells were acquired.

## Western blotting

To monitor expression of Vpu, transfected HEK293T cells were washed in PBS and lysed in Western Blot lysis buffer (150 mM NaCl, 50 mM HEPES, 5 mM EDTA, 0.1% NP40, 500 μM $Na_3VO_4$, 500 μM NaF, pH 7.5). Lysates were subsequently mixed with Protein Sample Loading Buffer supplemented

with 10% β-mercaptoethanol and heated at 95℃ for 5 min. Proteins were separated on NuPAGE 4–12% Bis-Tris Gels, blotted onto Immobilon-FL PVDF membranes and stained using primary antibodies directed against Vpu (CH293 and STCO1 Vpu: NIH AIDS Reagent Program, #11942, dilution 1:5000; CH077 Vpu: antiserum kindly provided by S. Bolduan, dilution 1:500), p24 (Abcam, #ab9071, dilution 1:5000), GAPDH (BioLegend, #607902, dilution 1:1000) and Infrared Dye labeled secondary antibodies (LI-COR IRDye, dilution 1:20000). Signal Enhancer HIKARI kit for Western Blotting and ELISA (Nacalai Tesque, #02267–41) was used for Vpu and p24 staining. Proteins were detected using a LI-COR Odyssey scanner.

## Virus release assay

To assess Vpu-mediated enhancement of virion release, HEK293T cells were seeded in 6-well plates and transfected with 5 µg of a proviral construct and increasing amounts of a plasmid coexpressing human tetherin. 40 hr post-transfection, infectious virus yield was determined by infecting TZM-bl reporter cells.

## NF-κB reporter assay

HEK293T cells were co-transfected with a firefly luciferase reporter construct under the control of three NF-κB binding sites (200 ng), a *Gaussia* luciferase construct under the control of a minimal pTAL promoter for normalization (25 ng), an expression plasmid for a constitutively active mutant of IKKβ (100 ng), and a proviral HIV-1 construct (100 ng). All transfections were performed in 96-well plates, in triplicates, using the calcium phosphate method. 40 hr post-transfection, firefly and *Gaussia* luciferase activities were determined using the Luciferase Assay System from Promega and a homemade assay, respectively. Firefly luciferase signals were normalized to the respective *Gaussia* luciferase control.

## Immunofluorescence microscopy

HEK293T cells were seeded on Poly-L-Lysine-coated coverslips in 24-well plates and transfected with a proviral construct or an empty vector (750 ng). Two days post-transfection, cells were stimulated with 5 ng/ml TNFα for 15 min or left untreated. Subsequently, cells were fixed for 20 min at room temperature with 4% paraformaldehyde and permeabilized using PBS containing 0.5% Triton X-100% and 5% FCS for 20 min at room temperature. Proteins were detected using antibodies against HIV-1 p24 (Abcam, #ab9071, dilution 1:500), p65 (Santa Cruz, #sc-372, dilution 1:150) and fluorophore-conjugated secondary antibodies (Thermo Scientific #A31571 and #A11011, dilution 1:350). Nuclei were visualized by DAPI staining. Cells were mounted in Mowiol mounting medium (Cold Spring Harbor Protocols) and analyzed using confocal microscopy (LSM 710, Zeiss) and the corresponding software (Zeiss Zen Software). The subcellular localization of p65 was quantified using Fiji. To this end, the average p65 intensity of defined areas ($25 \times 25$ squares) in the cytoplasm and nucleus of p24 positive cells was determined in a blinded manner. The p65 intensity in the nucleus was divided by that of the cytoplasm to determine the nuclear translocation rate of p65. All ratios were normalized to p65 levels of p24-negative bystander cells to account for possible differences in staining efficiency between different wells.

## Library construction and transcriptome sequencing

Total RNA was isolated and purified from infected CD4 +T cells using the RNeasy Plus Mini Kit (QIAGEN, Hilden, Germany). To monitor RNA quality, RNA Integrity Number (RIN) score was analyzed using an Agilent 2100 Bioanalyzer System (*Supplementary file 1*) (*Schroeder et al., 2006*). RNA sequencing libraries were generated from 150 ng of RNA using Illumina's TruSeq Stranded mRNA Sample Prep Kit following manufacturer's instructions, modifying the shear time to 5 min. RNA libraries were multiplexed and sequenced with 50 base pair (bp) single reads (SR50) to a depth of approximately 30 million reads per sample on an Illumina HiSeq4000.

## qRT-PCR

Total RNA was isolated and purified from pelleted CD4 +T cells using the RNeasy Plus Mini Kit (QIAGEN, Hilden, Germany). Cells were homogenized by vortexing for 30 s. Subsequent gDNA digestion was performed using the DNA-*free* DNA Removal Kit (ThermoFisher Scientific, Waltham, MA). The

maximal amount of RNA was reversely transcribed with the PrimeScript RT Reagent Kit (Perfect Real Time) (TAKARA, Kusatsu, Japan) using Oligo dT primers and random hexamers. cDNA was subjected to quantitative real time PCR using primer/probe sets for human IFNB1 and GAPDH (Thermo-Fisher Scientific, Waltham, MA) in singleplex reactions. Samples were analyzed in triplicates. Ct data were processed relative to the GAPDH control.

### Flow-cytometric cytokine quantification

Three days post-infection, cell culture supernatants of infected CD4 +T cells were analyzed using the LEGENDplex Human Anti-Virus Response Panel, (BioLegend, San Diego, CA, Cat#740390) according to the manufacturer's instructions. 5000 events were acquired per sample.

### Membrane-based cytokine array

Three days post-infection, cleared cell culture supernatants were analyzed using the RayBio C-Series Human Cytokine Antibody Array C5 (RayBiotech, Norcross, GA, Cat#AAH-CYT-5–8). The assay was performed according to the manufacturer's instructions with the exception that bound antibodies were labeled with IRDye 800CW labeled streptavidin (LI-COR Biosciences, Lincoln, NE, Cat#926–32230, dilution 1:1000) for detection via the Odyssey infrared imaging system (LI-COR Biosciences, Lincoln, NE). Spot intensities were quantified using LI-COR Image Studio Lite Version 3.1.

### Quantification and statistical analyses

Cells from four donors were mock infected or infected with three different HIV-1 isolates, each either wt or one of three *vpu* mutants (*Figure 1B*), resulting in a total number of 52 samples subjected to RNA sequencing and subsequent statistical analysis (*Supplementary file 1*). The analysis of three different primary HIV-1 isolates in four healthy donors (two women, two men) allowed us to reliably determine the relative impact of donor, virus and *vpu* status on host gene expression. Kallisto (v0.43) was used to build an index using the GRCh38 *Homo sapiens* reference transcriptome, followed by pseudo-alignment of RNA-Seq reads and quantification of transcript abundances (*Bray et al., 2016*). All bioinformatic and statistical analyses were performed in R (v3.3.2). Transcript counts were associated with gene IDs for gene-level summarization. No outliers were detected based on principal component analysis and considering Cook's distance (*Figure 2B* and *Figure 2—figure supplement 1C*). Differential expression analyses were performed using the DESeq2 package (v1.16.1) (*Love et al., 2014*), which is based on negative binomial generalized linear models. The GAGE package (v2.22.0) was used for all gene set and pathway analyses. It performs a two-sample t-test on log fold changes of gene expression levels and derives a global p-value using a meta-test on the negative log sum of p-values from all one-on-one comparisons (*Luo et al., 2009*). Statistical significance for all tests was defined as q < 0.1 after FDR correction. Phenotypic properties of the HIV-1 clones analyzed in this study (*Figure 1C–E* and *Figure 1—figure supplements 1B* and *2B*) and their effects on cytokine production (*Figure 4*) are shown a mean values ± standard error of the man (SEM). In these experiments, statistical significance was tested using repeated measure (RM) one-way ANOVA with Greenhouse-Geisser correction and Dunnett's multiple comparison test (*Figures 1C, D*, *4A and C*) or ordinary one-way ANOVA and Dunnett's multiple comparison test (*Figure 1—figure supplements 1B* and *2B*).

### Sequence analyses

Vpu amino acid sequences were aligned using MultAlin (*Corpet, 1988*) and the Vpu transmembrane domain was predicted using TMHMM Server v. 2.0 (*Krogh et al., 2001*).

## Availability of data, software, and research materials

The RNA-Seq data have been uploaded to the Gene Expression Omnibus (GEO) database (accession number #GSE117655).

## Acknowledgments

We thank Martha Mayer, Regina Linsenmeyer, Kerstin Regensburger, Susanne Engelhart and Daniela Krnavek for excellent technical assistance and Konstantin Sparrer for help with confocal imaging

analysis. We also thank Beatrice H. Hahn and Bernd Baumann for providing proviral HIV-1 clones and NF-κB reporter vectors, respectively. We further thank Birgit Liss and Desiree Spaich for providing access to the 2100 Bioanalyzer. TZM-bl cells were provided by the NIH AIDS Reagent Program, Division of AIDS, NIAID, NIH from John C Kappes, Xiaoyun Wu and Tranzyme Inc (*Platt et al., 1998*). Vpu-specific antisera were kindly provided by Beatrice H Hahn (via the NIH AIDS Reagent Program, Division of AIDS, NIAID, NIH) and Sebastian Bolduan. RNA-Seq was conducted at the IGM Genomics Center, University of California, San Diego, La Jolla, CA. SL, KH, and DH were supported by the International Graduate School in Molecular Medicine Ulm (IGradU).

## Additional information

### Funding

| Funder | Grant reference number | Author |
| --- | --- | --- |
| International Graduate School in Molecular Medicine Ulm | | Simon Langer<br>Kristina Hopfensperger<br>Dominik Hotter |
| Deutsche Forschungsgemeinschaft | 404687549 | Simon Langer |
| James B. Pendleton Charitable Trust | | Simon Langer<br>Paul D De Jesus<br>Kristina M Herbert<br>Lars Pache<br>Sumit K Chanda |
| European Research Council | | Johannes A van der Merwe<br>Frank Kirchhoff |
| Deutsche Forschungsgemeinschaft | SPP 1923 | Frank Kirchhoff |

The funders had no role in study design, data collection and interpretation, or the decision to submit the work for publication.

### Author contributions

Simon Langer, Conceptualization, Data curation, Formal analysis, Funding acquisition, Investigation, Writing—review and editing; Christian Hammer, Data curation, Software, Formal analysis, Visualization, Methodology, Writing—review and editing; Kristina Hopfensperger, Formal analysis, Validation, Investigation, Writing—review and editing; Lukas Klein, Formal analysis, Investigation; Dominik Hotter, Resources, Investigation, Methodology, Writing—review and editing; Paul D De Jesus, Lars Pache, Data curation, Formal analysis; Kristina M Herbert, Data curation, Formal analysis, Methodology, Writing—review and editing; Nikaïa Smith, Methodology, Writing—review and editing; Johannes A van der Merwe, Methodology; Sumit K Chanda, Resources, Funding acquisition, Writing—review and editing; Jacques Fellay, Resources, Software; Frank Kirchhoff, Resources, Supervision, Funding acquisition, Writing—review and editing; Daniel Sauter, Conceptualization, Resources, Data curation, Formal analysis, Supervision, Funding acquisition, Visualization, Methodology, Writing—original draft, Project administration

### Author ORCIDs

Christian Hammer ORCID http://orcid.org/0000-0003-4548-7548
Nikaïa Smith ORCID http://orcid.org/0000-0002-0202-612X
Jacques Fellay ORCID http://orcid.org/0000-0002-8240-939X
Daniel Sauter ORCID http://orcid.org/0000-0001-7665-0040

### Ethics

Human subjects: The use of human PBMCs was approved by the Ethics Committee of the Ulm University Medical Center (application #50/16). All donors were anonymized and provided informed written consent.

Decision letter and Author response
Decision letter https://doi.org/10.7554/eLife.41930.020
Author response https://doi.org/10.7554/eLife.41930.021

## Additional files

### Supplementary files

• Supplementary file 1. Identity and integrity of RNA-Seq samples.
DOI: https://doi.org/10.7554/eLife.41930.012

• Supplementary file 2. Custom-defined gene set of NF-κB target genes.
DOI: https://doi.org/10.7554/eLife.41930.013

• Supplementary file 3. Custom-defined gene set of host restriction factors.
DOI: https://doi.org/10.7554/eLife.41930.014

• Supplementary file 4. Primers used for mutagenesis of *vpu*
DOI: https://doi.org/10.7554/eLife.41930.015

• Transparent reporting form
DOI: https://doi.org/10.7554/eLife.41930.016

### Data availability

RNA sequencing data have been uploaded to the Gene Expression Omnibus (GEO) database (accession number GSE117655).

The following dataset was generated:

| Author(s) | Year | Dataset title | Dataset URL | Database and Identifier |
|---|---|---|---|---|
| Langer S | 2018 | HIV-1 Vpu exerts broad immunosuppressive effects by inhibiting NF-κB-dependent gene expression | https://www.ncbi.nlm.nih.gov/geo/query/acc.cgi?acc=GSE117655 | NCBI Gene Expression Omnibus, GSE117655 |

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
