## [Decision Letter]

Thank you for submitting your article "HIV-1 Vpu is a potent transcriptional suppressor of NF-κB-elicited antiviral immune responses" for consideration by *eLife*. Your article has been reviewed by three peer reviewers, and the evaluation has been overseen by a Reviewing Editor and Michel Nussenzweig as the Senior Editor. The following individuals involved in review of your submission have agreed to reveal their identity: Fabrizio Mammano (Reviewer #3).

The reviewers have discussed the reviews with one another and the Reviewing Editor has drafted this decision to help you prepare a revised submission.

Summary:

In this manuscript, Langer and colleagues investigate the role of HIV Vpu in modulating gene expression in human cells. Several groups have previously shown that HIV Vpu is able to modulate the antiviral immune response. The pathway through which this is accomplished, however, is the subject of a controversy. Some of the authors of the present manuscript had previously provided evidence that Vpu acts by inhibiting the NF-κB pathway, against the alternative model of inhibition by degradation of IRF-3. Data in this report lend strong additional support for the NF-κB pathway.

In addition, Vpu was previously shown to inhibit NF-κB by two independent mechanisms: by stabilizing the NF-κB inhibitor IκB, or by counteracting tetherin (in its role of sensor of virus infection). The main aim of the present study was to evaluate the relative contribution of these two mechanisms.

In order to determine how Vpu alters gene expression of infected cells, the authors infect primary CD4+ T cells from four different healthy donors (two males and two females) with three primary isolates with and without Vpu. In addition, they use specific Vpu mutants unable to stabilize IκB or unable to counteract tetherin to dissect the molecular mechanism behind these changes. By conducting RNA-Seq analyses of cells infected by WT HIV or mutants with disrupted or selectively mutated Vpu alleles, the authors show that Vpu preferentially suppresses the expression of genes involved in innate and adaptive immunity. This is in part due to its capacity to interfere with NF-κB downstream of tetherin, while impairing the capacity of Vpu to counteract tetherin did not result in any measurable difference in gene expression.

The authors clearly show that HIV Vpu inhibits antiviral immune response. Indeed, infection of CD4+ T cells with viruses lacking Vpu results in an increased expression of genes involved in innate and adaptive immunity. The presented data provide strong evidence that Vpu inhibition of immune response is dependent on its ability to block NF-κB activation downstream tetherin.

Lastly, the authors analyze the cytokines produced by CD4+ T cells during infection by both flow cytometry quantification as well as cytokine array. They observe an increase in the production of pro-inflammatory cytokines in cells infected with *Δvpu* virus or the mutant Vpu unable to stabilize IκB thus further confirming that Vpu down-regulates immune response.

Taken together this well-done study gives a global picture of the effect of HIV Vpu in primary human CD4+ T cell. It allows to evaluate the relative perturbation of gene expression and allows to directly compare the tetherin-mediated and tetherin-independent pathway. Overall, this study is well done and important for the field but some limitations should be addressed as outlined in the essential revisions section.

Essential revisions:

The strengths of this study are the unbiased analyses, which allow to directly compare different potential pathways. The identification of several genes whose expression is modulated by Vpu, which will trigger further exploration. For instance, the expression of DNA-sensors was found to be specifically suppressed by Vpu through interference with NF-κB. However, the following points require clarification.

A clear limitation is that the analysis was not performed on sorted infected cells, introducing a bias with respect to the level of infection, possibly resulting in an underestimation of the effects of Vpu. This limitation needs to be clearly stated and discussed.

The use of VSV-G pseudotyped viruses for the infections does not affect the comparison of the infections +/- Vpu, but likely will impact the overall gene expression pattern. This limitation needs to be clearly stated and discussed.

The impact of infection rate on gene expression (subsection “Modulation of CD4+ T cell RNA expression profiles by wild type HIV-1 infection) is expected but somewhat worrisome. Did the authors normalize Vpu +/- for each virus based on infectivity or simply on p24? A clear trend can be seen in Figure 3E, where the fold change between Vpu-stop and WT is generally higher for the most infectious virus (CH077) and lower for the less infectious (CH293), suggesting that the infectivity effect has not been neutralized.

The fact that the R45K mutation in the context of the CH077 Vpu did not provide a phenotype similar to the other mutants is perplexing (Figure 1). This may be due to differences in the primary sequence of this Vpu compared to the other ones, in a way that introducing the R45K mutation in CH077 Vpu causes defects in Vpu expression. Please, include western blots in Figure 1 to rule out any differences in expression between the wild type Vpu and their R45K/R50K mutants. In addition, to verifying the expression/stability of the R45K mutant by WB, please also measure activity/interaction in a different system from the one used in Figure 1C. The reviewers feel that determining nuclear translocation of NF-κB in HEK293T transfected with the three Vpus and their mutants by microscopy would be an appropriate approach.

Statistical tests used in Figure 1 seem inappropriate (2 by 2 comparisons), as they do not correct for multiple comparisons. It is important to verify that the difference shown in Figure 1C is statistically significant using a test that corrects for multiple comparisons (Kruskal-Wallis or Anova), because four Vpus were compared. If the mutant is not different from the WT in its IκB-related activity, then it should not be included in the comparisons between NF-κB-defective and WT Vpu (Figures 3 and 4).

---

## [Author Response]

Essential revisions:The strengths of this study are the unbiased analyses, which allow to directly compare different potential pathways. The identification of several genes whose expression is modulated by Vpu, which will trigger further exploration. For instance, the expression of DNA-sensors was found to be specifically suppressed by Vpu through interference with NF-κB. However, the following points require clarification.A clear limitation is that the analysis was not performed on sorted infected cells, introducing a bias with respect to the level of infection, possibly resulting in an underestimation of the effects of Vpu. This limitation needs to be clearly stated and discussed.

We now discuss a possible bias and underestimation of Vpu effects due to the use of unsorted cells:

“Furthermore, the use of unsorted cells does not allow to discriminate between effects in infected and uninfected cells and may introduce a bias with respect to infection rates. […] In fact, the present approach may even underestimate the inhibitory effects of Vpu due to the presence of 25-80% uninfected bystander cells and the pre-activation of CD4+ T cells with PHA.”

The use of VSV-G pseudotyped viruses for the infections does not affect the comparison of the infections +/- Vpu, but likely will impact the overall gene expression pattern. This limitation needs to be clearly stated and discussed.

We agree and discuss a potential impact of pseudotyping:

“One limitation of the present study is the use of VSV-G for pseudotyping, which may influence sensing and overall gene expression.”

The impact of infection rate on gene expression (subsection “Modulation of CD4+ T cell RNA expression profiles by wild type HIV-1 infection) is expected but somewhat worrisome. Did the authors normalize Vpu +/- for each virus based on infectivity or simply on p24? A clear trend can be seen in Figure 3E, where the fold change between Vpu-stop and WT is generally higher for the most infectious virus (CH077) and lower for the less infectious (CH293), suggesting that the infectivity effect has not been neutralized.

For each of the three HIV-1 strains, wild type and mutant viruses were adjusted for their infectivity in primary CD4+ T cells, not for their p24 levels (subsection “Modulation of CD4+ T cell RNA expression profiles by wild type HIV-1 infection”, second paragraph and subsection “Generation of virus stocks”). In fact, the results shown in Figure 3E are in agreement with a neutralized infectivity effect. Since CH077 (wt and mutant) infected more cells than CH293 (wt and mutant), the former induces antiviral immune signaling in a larger number of cells and therefore results in a stronger fold change upon loss of Vpu.

The fact that the R45K mutation in the context of the CH077 Vpu did not provide a phenotype similar to the other mutants is perplexing (Figure 1). This may be due to differences in the primary sequence of this Vpu compared to the other ones, in a way that introducing the R45K mutation in CH077 Vpu causes defects in Vpu expression. Please, include western blots in Figure 1 to rule out any differences in expression between the wild type Vpu and their R45K/R50K mutants.

Western blots of transfected HEK293T cells show that the R45K mutation does not decrease expression of CH077 Vpu (new Figure 1F). Furthermore, the finding that the R45K/R50K mutants counteract tetherin as efficiently as the respective wild type viruses (Figure 1E and Figure 1—figure supplement 2B) excludes a general defect of the R/K mutants.

In addition, to verifying the expression/stability of the R45K mutant by WB, please also measure activity/interaction in a different system from the one used in Figure 1C. The reviewers feel that determining nuclear translocation of NF-κB in HEK293T transfected with the three Vpus and their mutants by microscopy would be an appropriate approach.

As suggested by the reviewers, we monitored nuclear translocation of p65 in HEK293T cells transfected with the three wild type proviral constructs or their vpu mutants (new Figure 1—figure supplement 1B, subsection “Generation of selective Vpu mutants”).

Statistical tests used in Figure 1 seem inappropriate (2 by 2 comparisons), as they do not correct for multiple comparisons. It is important to verify that the difference shown in Figure 1C is statistically significant using a test that corrects for multiple comparisons (Kruskal-Wallis or Anova), because four Vpus were compared. If the mutant is not different from the WT in its IκB-related activity, then it should not be included in the comparisons between NF-κB-defective and WT Vpu (Figures 3 and 4).

To correct for multiple comparisons, we now use ANOVA for the data shown in Figure 1C, D, Figure 1—figure supplement 1B, 2B and Figure 4A-C and changed the Materials and methods section accordingly (subsection “Quantification and statistical analyses”).

Although introduction of R45K in CH077 did not affect the nuclear translocation of p65 (Figure 1—figure supplement 1B), it significantly reduced the activation of NF-κB in the luciferase reporter assay (Figure 1C). While the reporter assay directly monitors NF-κB-driven gene expression, the levels of p65 in the nucleus do not necessarily reflect overall NF-κB activity since nuclear p65 may still be inactive and since NF-κB molecules lacking p65 also exist. We therefore decided to include the CH077 R45K mutant in Figures 3 and 4. Importantly, the R/K mutations significantly increase antiviral immune signaling, even if we include the CH077 R45K mutant showing only a partial loss of function (Figure 3D). If anything, inclusion of this mutant underestimates the effects of Vpu mediated NF-κB inhibition.